# Inhibition of Human Platelet Aggregation and Low-Density Lipoprotein Oxidation by *Premna foetida* Extract and Its Major Compounds

**DOI:** 10.3390/molecules24081469

**Published:** 2019-04-13

**Authors:** Roza Dianita, Ibrahim Jantan

**Affiliations:** 1School of Pharmaceutical Sciences, Universiti Sains Malaysia, 11800 Pulau Pinang, Malaysia; 2School of Pharmacy, Faculty of Health and Medical Sciences, Taylor’s University, Lakeside Campus, 47500 Subang Jaya, Selangor, Malaysia

**Keywords:** *Premna foetida*, anti-LDL oxidation, antiplatelet aggregation, flavonoids, quercetin, apigenin

## Abstract

Many *Premna* species have been used in traditional medicine to treat hypertension and cardiac insufficiency, and as a tonic for cardiac-related problems. Some have been reported to possess cardiovascular protective activity through several possible mechanisms, but not *Premna foetida*. In the present study, the methanol extract of *P. foetida* leaves (PFM) and its isolated compounds were evaluated for their ability to inhibit copper-mediated human low-density lipoprotein (LDL) oxidation and arachidonic acid (AA)- and adenosine diphosphate (ADP)-induced platelet aggregation. Six flavonoids, three triterpenoids, vanillic acid and stigmasterol were successfully isolated from PFM. Of the isolated compounds, quercetin was the most active against LDL oxidation (IC_50_ 4.25 µM). The flavonols were more active than the flavones against LDL oxidation, suggesting that hydroxyl group at C-3 and the catechol moiety at B-ring may play important roles in protecting LDL from oxidation. Most tested flavonoids showed stronger inhibition towards AA-induced than the ADP-induced platelet aggregation with apigenin exhibiting the strongest effect (IC_50_ 52.3 and 127.4 µM, respectively) while quercetin and kaempferol showed moderate activity. The results suggested that flavonoids, especially quercetin, apigenin and kaempferol were among the major constituents of *P. foetida* responsible for anti-LDL oxidation and anti-platelet aggregation.

## 1. Introduction

Herbs have been widely used as food supplements, flavors, and in traditional medicine, which directly and indirectly benefit the body against cardiovascular risk conditions such as hyperlipidemia or hypercholesterolemia. The cardiovascular-protective activity from these plants is often derived through interventions in lipid metabolism and/or protein regulation of the vascular and related homeostasis systems. The plants could be acting as lipid lowering agents, antioxidants, especially towards low-density lipoproteins (LDL), regulator of biochemicals and immunochemicals contributing to atherogenesis, and as antithrombotic agents [1,2]. LDL is a unique molecule that can fuel the atherogenosis process once it has gone through oxidation or modification process within the arterial wall due to chronic immune-inflammation [3]. The oxidized LDLs (Ox-LDLs) are easily taken up by the macrophages which engulf the lipids (especially cholesterol) including the ox-LDL and convert the macrophage into lipid-laden foam cells. Ox-LDL also attracts the monocytes and induces more inflammation and endothelial dysfunction, contributing to the more advanced atherosclerotic lesion stage [4].

Many epidemiological and biological studies have shown polyphenolic-rich plant extracts can act as antioxidant, vasodilator, anti-inflammatory, antiatherogenic, antithrombotic and apoptosis agents [5]. Phytochemicals such as flavonoids, phenolic acids and sterols exert their cardioprotective effects through numerous mechanisms, including scavenging of reactive oxygen species (ROS) and enhancement of antioxidant activity, chelating metal ions, improving lipid profiles via inhibition of key enzymes in lipid biosynthesis and metabolism, regulating biomarkers in endothelial dysfunction/injury such as nitric oxide (NO) and leukocyte adherence, and inhibiting the proliferation of smooth muscle cells [6,7]. Although the antioxidant activity of many plants has been well demonstrated, direct evidence of acute therapeutic benefits of plant extracts and their phenolic compounds in cardiovascular disorders remains sparse and data on anti-LDL oxidation have been few.

*Premna foetida* Reinw. Ex Blume is used in traditional medicine to treat fever [8], cough [8,9], constipation, liver and spleen problems, and as an antimalarial [10]. The leaves are used as a salad or as an ingredient in curries [11]. The closely related species, *P. serratifolia* has been used in traditional medicine to treat hypertension and cardiac insufficiency [12], and as a tonic for cardiac-related problems [10]. There is little information however on the phytochemicals and biological activities of *P. foetida*, although other species such as *P. mucronata* and *P. serratifolia*, have been reported to have protective activity towards cardiovascular diseases through several mechanisms [13,14,15,16]. Thus, our present study aimed to investigate the secondary metabolites of *P. foetida* leaves as well as to evaluate their anti-LDL oxidation and antiplatelet aggregation activities.

## 2. Results and Discussion

### 2.1. HPLC Analysis

A validated HPLC method for analysis of *P. foetida* extract was developed on a reversed-phase (C-18) column with a gradient system of water-acetonitrile as mobile phase. The HPLC method was validated for its accuracy, specificity, linearity and LOD-LOQ parameters (Table 1). For this purpose, quercetin was chosen as an external standard to validate the method. The results show the method has good specificity, indicated by the percentage of recovery which fell within the 90–110% range. The small values of the standard deviation (SD) of the retention times (RT), the consistency of the responses (AUC, area under curve) vs concentration in intra- and inter-day experiments, and the good linearity (*R*^2^ = 0.998) of the calibration curve (y = 1760.1x − 54930), indicated the HPLC method has a good precision and linearity. The LOD and LOQ values were calculated as 5.62 and 17.04 ng/mL, respectively. The HPLC chromatogram of the methanol extract of *P. foetida* (Figure 1) shows the presence of acaciin, quercetin, apigenin and kaempferol at 15.33, 30.19, 33.95 and 35.25 min, respectively based on retention time comparison of standard compounds and spiking technique. The extract was found to contain quercetin at 314.6 µg/mL while acaciin, apigenin and kaempferol were simultaneously quantified at 520.1, 19.0 and 47.1 µg/mL, respectively.

### 2.2. Isolation and Identification of Compounds

Eleven known compounds, comprising of six flavonoids and their glycosides were successfully isolated from the methanol extract of the leaves of *P. foetida* along with three triterpenoids, stigmasterol and vanillic acid (Figure 2).

The compounds were isolated from the extract by various chromatographic techniques. The structures of the compounds, acaciin (**1**), quercitrin (**2**), ursolic acid (**3**), stigmasterol (**4**), maslinic acid (**5**), corosolic acid (**6**), acacetin (**7**), apigenin (**8**), kaempferol (**9**), quercetin (**10**) and vanillic acid (**11**), were elucidated by a combination of HRMS and NMR techniques, and also by comparison of their spectroscopic and physicochemical data with literature values [17,18,19,20,21,22,23,24]. This is the first report on the phytochemicals isolated from *P. foetida*. The presence of acacetin, quercetin, apigenin stigmasterol, ursolic acid, and vanillic acid in other species of *Premna* has been reported previously [25].

### 2.3. LDL Antioxidant Activity

Table 2 summarizes the percent inhibitions against LDL oxidation and the IC_50_ values of fractions and identified compounds from *P. foetida* with probucol (LDL oxidation IC_50_ value 0.40 µM) as the positive control. 

Probucol is a cholesterol-lowering drug which works by increasing the rate of LDL catabolism and at the same time, providing oxidation resistance towards LDL [26,27]. Inversely, probucol caused reduction in HDL [28,29] which overrides its anti-atherosclerotic effects. Thus, this drug is no longer the first line drug to lower cholesterol level and has been replaced by statins. All fractions displayed antioxidant activity towards LDL oxidation in a concentration-dependent manner. At high concentration (25 µg/mL), the MeOH fraction of *P. foetida* leaves showed the highest inhibitory activity (80.55%) followed by CHCl_3_ and *n*-hexane fractions (71.35 and 59.80%, respectively). However, at low concentration (3.13 µg/mL), the CHCl_3_ fraction produced more inhibitory activity than the MeOH fraction. Referring to Figure 3, a concentration-vs-inhibition relationship is observed and the slope trend exerted by the MeOH fraction is steeper than that of the CHCl_3_ fraction. This resulted in lower IC_50_ value of the CHCl_3_ fraction (3.95 µg/mL) than the MeOH fraction (5.77 µg/mL). Most of the flavonoids isolated in this study were obtained from the CHCl_3_ fraction (Figure 4). Thus it is deduced that the rich aglycon-flavonoid content present in the fraction might be responsible for such phenomenon.

Nine compounds isolated from *P. foetida* were evaluated for their inhibition activity against LDL oxidation and platelet aggregation. Two isolated triterpenes, corosolic acid and maslinic acid, were not included in the assays due to very low yield. Among the six isolated flavones, quercetin was the most active inhibitor towards LDL oxidation with an IC_50_ value of 4.25 µM, followed by kaempferol, quercitrin, acaciin, apigenin, and acacetin (8.02, 9.94, 12.51, 15.11 and 17.05 µM, respectively). Vanillic acid, ursolic acid, and stigmasterol moderately inhibited LDL oxidation with IC_50_ values of 59.99, 73.85 and 79.60 µM, respectively. Apparently, those isolated compounds were among the compounds responsible for the inhibitory activity of LDL oxidation by the extract/fractions of *P. foetida*.

Oxidation of LDL by Cu^2+^ ion occurs at low concentration. Metal Cu^2+^ ion would bind to specific amino acid residues in apoB (i.e histidine-containing sites on apoB-100) initiating the oxidation of polyunsaturated fatty acyl chains in the core of LDL molecules yielding aldehydes such as malondialdehyde (MDA) as oxidation products [30]. This oxidation process is broken down into three stages which are initiation, propagation and decomposition or degradation [31]. The initiation stage, also known as lag phase, is the phase where the oxidation is suppressed by the presence of endogenous antioxidants in the LDL particle, indicated by low level of lipid peroxidation products. This is followed by a rapid propagation phase where the antioxidants are rapidly consumed and there is a high level of lipid peroxide products. Finally, the double bonds are cleaved and aldehydes formed. Other changes on LDL during oxidation included alteration of protein moieties and fragmentation of apoB due to oxidative scission. Thus, exogenous antioxidants might be introduced in the initiation or during the propagation stages to scavenge the aldehydes and stop further damage. Our findings suggest that flavonoids and other polyphenolic compounds were mostly responsible for the antioxidant activity of *P. foetida* towards LDL oxidation by one of more mechanism such as: (i) by decreasing the copper-binding sites on the LDL thus preventing it from oxidizing the LDL; and also (ii) by directly neutralizing radicals/aldehydes during propagation stage. However, further studies need to be carried out to further elucidate the mechanism of these compounds in protecting the LDL from oxidation, in vitro and in vivo.

Based on the IC_50_ values, the flavonols (kaempferol, quercetin, quercitrin) were more active against the LDL oxidation than the flavones (acacetin, acaciin, apigenin). It was probably due to the presence of hydroxyl group at C-3, which plays an important role in protecting the LDL from further oxidation. Glycosylation of the hydroxyl at C-3 reduces the ability of the flavonol to inhibit LDL oxidation, as seen in quercitrin in comparison to kaempferol. Yet, their inhibitory activities were still superior to the flavones (acacetin, acaciin, apigenin) which do not have hydroxyl substitution at C-3. Methylation of the hydroxyl group at C-4′ of the flavone (apigenin vs acacetin) also caused the antioxidant activity to decrease. Meanwhile, glycosylation of hydroxyl group at C-7 in acaciin apparently increased the antioxidant activity compared to acacetin and apigenin. It seems that glycosylation of 7-OH enhanced greater antioxidant effect in the flavone than in the unchanged 4′-OH. The different number and position of attached hydroxyl groups seems to play an important role in enhancing the anti-LDL oxidation activity of the flavonoids. Within flavonols (kaempferol, quercetin), *ortho*-substitution of hydroxyl groups at C-3′ and C-4′ of B-ring was crucial to increase the antioxidant activity. Our findings were in agreement with previous studies [32,33,34] that reported the structure-activity relationship of some flavonoids towards LDL oxidation. Hence, the results indicated that: (i) more free hydroxyl groups would likely result in higher antioxidant activity; (ii) two adjacent hydroxyl groups (a catechol moiety) on the B ring exerted superior antioxidant effect compared to the double bond and carbonyl groups on the C ring; (iii) a free hydroxyl group at C-3 and C-7 of the C ring contributed to the antioxidant activity with free 3-OH providing better antioxidant capacity than the 7-OH. Glycosylation of 7-OH might improve the antioxidant activity of the flavone, while glycosylation of 3-OH brought the opposite effect; and (iv) the C2-C3 double bond and hydroxyl group at C-4′ also contribute to the antioxidant effect.

### 2.4. Antiplatelet Activity

Table 3 shows the dose-dependent inhibition of platelet aggregation induced by AA and ADP by selected compounds from *P. foetida*. In general, most tested flavonoids showed better inhibition towards AA-induced platelet aggregation than ADP-induction. As a well-known irreversible competitor of COX-1, aspirin demonstrated strong inhibition (100%) towards AA-induced aggregation at 25 µg/mL (IC_50_ at 26.0 µM), but only showed moderate-to-weak activity against ADP-induced aggregation. Apigenin, kaempferol and quercetin also showed strong inhibition against AA-induced platelet aggregation at 100 µg/mL. Interestingly, when ADP was used as inducer, strong activity was only displayed by apigenin and quercetin, while kaempferol showed moderate platelet aggregation inhibition.

The antiplatelet activities of compounds from *P. foetida* (kaemperol, apigenin, and quercetin) were evaluated based on their inhibition of platelet aggregation in blood. Two inducers, arachidonic acid (AA, 0.5 µM) and adenosine diphosphate (ADP, 10 µM), were used. The assay used impedance to measure the electrode resistance of the adhered platelet aggregates. AA and ADP were used as platelet agonists to induce platelet aggregation. In the body, upon phospholipase A2 activation, AA is released from the platelets and converted into prostaglandins PGG_2_ and PGH_2_ by COX-1 which are further converted into TxA_2_ by thromboxane synthase. The PGH_2_ and TxA_2_ then interact with receptors on the platelet surface causing aggregation. Meanwhile, ADP is secreted from the dense platelet granules of platelets once the platelets are activated or there are cells damaged due to vascular injury. This activation is mediated by G-protein-coupled receptors, P_2_Y_1_ (Gq) and P_2_Y_12_ (Gi_2_), causing changes in platelet shape and platelet aggregation by expressing GP IIb-IIIa surface glycoprotein [35,36]. We assumed apigenin, quercetin and kaempferol not only competitively blocked the thromboxane enzyme that is responsible for transforming PGH_2_ into TxA_2_, but also inhibiting the activation of TxA_2_ receptor [37]. Additionally, quercetin might also bind to the subdomain IIA (site 1) of the human serum albumin [38], hence inhibiting platelet aggregation.

The structure-activity analysis indicated that the absence of substitution at C-3 was important to enhance antiplatelet activity. A single hydroxyl substitution at C-4′ provided better activity than the di-substitution of hydroxyl at position C-3′ and C-4′. However, hydroxyl trisubstitution at positions C-3′, C-4′ and C-5′ of the B ring greatly reduced the antiplatelet activity. Similarly, Beretz et al. [39] found the decrease of inhibitory effect of flavonoids towards platelet aggregation occurred with saturation of the C2-C3 double bond, lack of a carbonyl carbon at C-4, glycosylation of C-3, and increasing number of hydroxyl substituents. On the contrary, hydroxylation of the B ring either as a catechol group or a single hydroxyl at C-4′ [40], methylated B-ring and a non-hydroxylated planar C ring [41] potentially inhibit platelet function. Thus, further research (in silico or molecular in vitro studies) is required to study the relationship between those moieties and the molecular mechanisms.

Platelets and oxLDL are implicated in the pathogenesis of atheromateous lesions in the plaque which play a key role in acute arterial thrombosis. oxLDL contribute to atherosclerotic plaque destabilization and thrombus formation by promoting macrophages to stimulate tissue factor expression, a potent pro-coagulant, as well as matrix metalloproteinase-9 (MMP-9). Both tissue factor and MMP-9 are involved in plaque rupture and thrombotic cascade, which eventually promotes thrombosis [42]. Platelets adhered due to exposed collagen during endothelial cell injury or thrombosis thus activate the expression of various materials, including aggregation inducers such as ADP, AA and thrombin. The activation causes more platelets to aggregate on the (original) injured endothelial cell leading to plaque formation [43].

## 3. Materials and Methods

### 3.1. Chemicals and Reagents

Solvents and orthophosphoric acid for HPLC analysis were purchased from Fisher Scientific (Kuala Lumpur, Malaysia) while solvents for isolation work and NMR experiments were purchased from Merck Millipore (Darmstadt, Germany). Standards for HPLC analysis were obtained from Sigma-Aldrich (St. Louis, MO, USA). Silica gels and TLC plates used for isolation works were provided by Merck Millipore, while Sephadex^®^ LH-20 was from GE Healthcare (Little Chalfont, UK). Separation of plasma lipoprotein was carried out by using an Optima™ L-80 XP ultracentrifuge with Type 70.1 Ti (fixed angle, r 40.5–82.0 mm) rotor (Beckman Coulter, Fullerton, CA, USA). The platelet aggregation was measured by using a Whole Blood Aggregometer model 592 (Chrono-Log Corp., Havertown, PA, USA). The NMR data were gathered by using a Varian VnmrS 500 MHz instrument (Varian Inc., Palo Alto, CA, USA) and the NMR data were read by using ACD/NMR Processor Academic Edition (ACD/Labs, Toronto, ON, Canada). The MS data were measured on a MicroTOF-Q system (Bruker Daltonics, Santa Clara, CA, USA).

### 3.2. Plant Collection

The leaves of *Premna foetida* were collected from Batu Feringgi (Pulau Pinang, Malaysia) in May 2013. The plant was identified by a botanist at the Institute of Bioscience, Universiti Putra Malaysia while the specimen was deposited at the Herbarium Universiti Kebangsaan Malaysia, Bangi (voucher no. UKM-30011). No specific permissions were required for the collection of the plant samples and the collection of the plant samples did not involve endangered or protected species. The plant materials were then air-dried with no direct exposure to the sunlight followed by coarse grinding and then kept in airtight containers at room temperature prior to the extraction process.

### 3.3. Plant Sample Preparation for HPLC Analysis

About 10 g of air-dried and coarsely ground leaves of *P. foetida* were extracted with 250 mL of MeOH with continuous stirring for 8 h, followed by soaking for the next 16 h. The filtrate was then concentrated to remove the solvent residue, yielding a crude methanol extract. The extract stock solution (20 mg/mL) for HPLC was prepared in MeOH and filtered through 0.45 µm membrane filter. Standard’s stock solution was prepared at 1 mg/mL and further diluted into a series of 500, 250, 125, 62.5, 31.25 and 15.13 µg/mL.

### 3.4. HPLC Instrumentation and Analysis

The HPLC analysis was carried out on a 2535 Quaternary Gradient Module pump with 2998 Photodiode Array detector and analytical XBridge C18-column (4.6 × 250 mm, 5 µm); all from Waters (Milford, MA, USA). The data acquisition was performed by using Empower3 software. The mobile system consisted of solvent A (0.1% orthophosphoric acid) and solvent B (acetonitrile). Initially, A:B was 85:15. The ratio of solvent B was gradually increased to 85% at min 55 and 100% at min 60, which was maintained until 75 min. The curve was fixed at 7. The flow rate was maintained at 1 mL/min and volume of injection was 20 µL. The detection was measured at 254 and 360 nm. The method was validated based on the ICH Guidelines for validation of analytical procedure [44].

### 3.5. Isolation Procedure

The air-dried, coarsely ground leaves of *P. foetida* (1 kg) were macerated at room temperature, consecutively with *n*-hexane, CHCl_3_ and MeOH at a ratio 1:6 for 3 × 3 days for each solvent. The filtered extracts were concentrated by using rotary evaporator to obtain crude extracts (*n*-hexane, 15.06 g; CHCl_3_ 15.57 g; and MeOH 155.22 g). Part of the crude MeOH extract (90.4 g) was further fractionated with *n*-BuOH (3 × 500 mL). Acaciin (**1**, 15.97 mg) precipitated as opaque (off-white) plates/solids when the *n*-BuOH fraction was concentrated. The BuOH fraction (13.53 g) was further chromatographed on Sephadex LH-20 using an isocratic solvent system (MeOH) at a flow rate 20–25 mL per 5 min. Upon solvent evaporation, needle-like crystals were formed in vials 29–42. Further purification gave quercitrin (**2**, 7.73 mg). The CHCl_3_ extract (13 g) was subjected to vacuum liquid chromatography (VLC) on silica gel 60 (type H) and a gradient solvent system (hexane-EtOAc 10:0 to 0:10, followed by EtOAc-MeOH (5:5) to obtain 12 subfractions. Subfraction C4 (967.2 mg) was chromatographed using normal column chromatography on silica gel 60 with a gradient solvent system (hexane-EtOAc 9:1 to 0:10, EtOAc-MeOH 8:2 and 0:10) to yield ursolic acid (**3**, 5.2 mg) and stigmasterol (**4**, 4.4 mg). Subfraction C8 (316.2 mg) was chromatographed on silica gel 60 using a gradient solvent system (CHCl_3_/MeOH 10:0 to 4:6) to yield 14 subfractions. Further chromatography of these subfractions led to the isolation of maslinic acid (**5**, 0.9 mg), corosolic acid (**6**, 1.3 mg), acacetin (**7**, 16 mg) and apigenin (**8**, 24.3 mg). Subfraction C10 (4.12 g) was chromatographed on Sephadex LH-20 with isocratic CHCl_3_-MeOH (1:1) as solvent system to give kaempferol (**9**, 10.6 mg), quercetin (**10**, 15.1 mg) and vanillic acid (**11**, 12.7 mg). The structure elucidation and identification of the isolated compounds were carried out using the MS and NMR data as well as by comparison to the literature values. The NMR spectra are supplied in the Appendix A.

*Acacetin-7-O-diglucoside (Acaciin/Linarin)* (**1**)*:* opaque (off white) palette/solid; HRMS *m*/*z* 592.2425; ^1^H-NMR (DMSO-d_6_) ppm: δ_H_ 8.04 (2H, d, *J* = 9.5 Hz), 7.14 (2H, d, *J* = 9 Hz), 6.94 (1H, s), 6.78 (1H, d, *J* = 2 Hz), 6.44 (1H, d, *J* = 2 Hz), 5.05 (1H, d, *J* = 7 Hz), 4.53 (1H, s), 3.85 (3H, s), 3.84 (1H, d, *J* = 8.5 Hz), 3.42 (1H, d, *J* = 4.5 Hz), 1.06 (3H, d, *J* = 6 Hz); ^13^C-NMR (DMSO-d_6_) ppm: δ_C_ 182.51, 164.40, 163.40, 162.89, 161.59, 157.44, 128.94 (2), 123.13, 115.18 (2), 105.91, 104.27, 100.97, 100.35, 100.10, 95.22, 76.69, 76.10, 73.51, 72.49, 71.18, 70.81, 70.03, 68.78, 66.54, 56.04, 18.28.

*Quercetin-3-β-O-rhamnoside (Quercitrin)* (**2**) *:* pale yellow needles; HRMS *m*/*z* 433.1128; ^1^H-NMR (CD_3_OD) ppm: δ_H_ aglycon: 7.33 (1H, s), 7.30 (1H, d, *J* = 8.5 Hz), 6.91 (1H, d, *J* = 8.5 Hz), 6.37 (1H, s), 6.20 (1H, s), sugar moieties: 5.34 (1H, s), 4.21 (1H, s), 3.74 (1H, dd, *J* = 3, 9.5 Hz), 3.42 (1H, m), 3.33 (1H, overlap), 0.94 (3H, d, *J* = 6 Hz); ^13^C-NMR (CD_3_OD) ppm: δ_C_ aglycon: 178.23, 164.51, 161.80, 157.90, 157.11, 148.39, 145.00, 134.81, 121.53, 121.43, 115.49, 114.94, 104.45, 98.39, 93.29, sugar moeities: 102.53, 71.82–70.48 (4C), 16.23.

*Ursolic acid* (**3**): white amorphous solid; HRMS *m*/*z* 458.3747; ^1^H-NMR (CDCl_3_) ppm: δ_H_ 5.22 (1H, t, *J* = 3.5 Hz), 4.51 (2H, brs), 3.15 (1H, dd, *J* = 3.5, 4.5, 11 Hz), 3.20 (1H, d, *J* = 12 Hz), 2.10 (1H, dt, *J* = 4.5, 13 Hz), 1.93 (3H, brdd, *J* = 3.5, 9.8 Hz), 1.00–1.65 (overlap), 0.97 (3H, s), 0.96 (3H, s), 0.95 (3H, s), 0.88 (3H, d, *J* = 6 Hz), 0.85 (3H, s), 0.77 (3H, s); ^13^C-NMR (CDCl_3_) ppm: δ_C_ 180.17, 138.34, 125.31, 78.32, 55.33, 53.02, 48.28, 48.00, 41.86, 39.37, 39.09, 38.99, 38.60, 38.44, 36.76, 36.70, 32.95, 30.46, 27.87, 27.41, 26.49, 22.97, 20.26, 18.08, 16.49, 16.31, 15.01, 14.68.

*Stigmasterol* (**4**): colorless needless; HRMS *m*/*z* 395.3654 and 413.2640; ^1^H-NMR (CDCl_3_) ppm: δ_H_ 5.49 (1H, brs), 5.29 (1H, dd, *J* = 9 Hz), 5.15 (1H, dd, *J* = 9 Hz), 3.66 (1H, m), 1.15 (3H, s), 1.06 (3H, d, *J* = 6.5 Hz), 0.98 (3H, m), 0.94 (6H, d, *J* = 8 Hz), 0.84 (3H, s); ^13^C-NMR (CDCl_3_) ppm: δ_C_ 140.76, 138.32, 129.29, 121.71, 71.81, 56.88, 55.97, 50.70, 42.30 (2), 40.50, 39.69, 36.52, 31.91 (2), 31.66, 28.93, 27.27, 25.42, 24.38, 21.20, 21.08, 19.05, 18.99, 12.26, 12.06.

*Maslinic acid* (**5**): white amorphous gum; HRMS *m*/*z* 249.1536 and 413.2570; ^1^H-NMR (CDCl_3_) ppm: δ_H_ 5.25 (1H, m), 3.16 (1H, m), 2.90 (1H, dd), 2.02 (1H, dd), 1.19 (3H, s), 1.16 (3H, s), 0.94 (3H, s), 0.91 (3H, s), 0.84 (3H, s), 0.80 and 0.81 (6H, d).

*Corosolic acid* (**6**): colorless, amorphous gum; HRMS *m*/*z* 472.3540; ^1^H-NMR (CDCl_3_) ppm: δ_H_ 5.28 (1H), 3.67 (1H, m), 2.98 (1H, d, *J* = 9.5 Hz), 2.04 (1H, m), 2.03 (1H, m), 1.13 (3H), 1.01 (3H), 0.97 (3H), 0.92 (3H), 0.89 (3H), 0.81 (3H), 0.76 (3H); ^13^C-NMR (CDCl_3_) ppm: δ_C_ 180.66, 143.87, 122.00, 83.43, 68.51, 55.20, 47.52, 46.29, 46.15, 45.87, 41.69, 41.13, 39.21, 39.10, 38.08, 33.79, 32.94, 32.53, 32.46, 30.58, 28.44, 27.54, 25.75, 23.42, 23.35, 22.93, 18.26, 16.74, 16.59, 16.40.

*Acacetin* (**7**): yellow needles; HRMS *m*/*z* 284.0681; ^1^H-NMR (CDCl_3_) ppm: δ_H_ 7.85 (2H, dd, *J* = 2, 7 Hz), 7.01 (2H, dd, *J* = 2, 6.5, 8.5 Hz), 6.54 (1H, s), 6.24 (1H, s), 6.43 (1H, s); ^13^C-NMR (CDCl_3_) ppm: δ_C_ 182.56, 164.43, 164.37, 162.77, 161.61, 158.01, 128.02 (2C), 123.31, 114.39 (2C), 104.37, 103.03, 99.20, 94.17, 55.17.

*Apigenin* (**8**): yellow powder; HRMS *m*/*z* 270.0525; ^1^H-NMR (DMSO-d_6_) ppm: δ_H_ 6.17 (1H, d, *J* = 2 Hz), 6.46 (1H, d, *J* = 2 Hz), 6.77 (1H, s), 6.90 (2H, d, *J* = 9 Hz), 7.91 (2H, d, *J* = 8.5 Hz), 12.95 (1H, s, OH), 10.82 (1H, brs, OH), 10.35 (1H, brs, OH); ^13^C-NMR (DMSO-d_6_) ppm: δ_C_ 182.20, 164.58, 164.17, 161.89, 161.61, 157.75, 128.93, 121.60, 116.40, 104.13, 103.28, 99.27, 94.41.

*Kaempferol* (**9**): yellow powder; HRMS *m*/*z* 286.0474; ^1^H-NMR (CD_3_OD) ppm: δ_H_ 8.06 (2H, d, *J* = 8.5 Hz), 6.88 (2H, d, *J* = 8 Hz), 6.36 (1H, s), 6.16 (1H, s); ^13^C-NMR (CD_3_OD) ppm: δ_C_ 175.89, 164.13, 161.05, 159.09, 156.79, 146.57, 135.69, 129.25, 122.30, 114.86, 103.10, 97.83, 93.04.

*Quercetin* (**10**): pale yellow powder; HRMS *m*/*z* 302.0423; ^1^H-NMR (DMSO-d_6_) ppm: δ_H_ 7.66 (1H, d, *J* = 2 Hz), 7.52 (1H, dd, *J* = 1.5, 2, 8.5 Hz), 6.86 (1H, d, *J* = 9 Hz), 6.38 (1H, s), 6.16 (1H, s), 12.48 (1H, s, OH), 10.77 (1H, brs, OH), 9.58 (1H, brs, OH), 9.36 & 9.30 (2H, brs each); ^13^C-NMR (DMSO-d_6_) ppm: δ_C_ 176.27, 164.31, 161.15, 156.56, 148.13, 147.21, 145.48, 136.17, 122.37, 120.40, 116.03, 115.48, 103.43, 98.60, 93.78.

*Vanillic acid* (**11**): white needles; HRMS *m*/*z* 168.042; ^1^H-NMR (CD_3_OD) ppm: δ_H_ 7.55 (2H, overlap), 6.83 (1H, d, *J* = 8.5 Hz), 3.89 (3H, s, OCH_3_); ^13^C-NMR (CD_3_OD) ppm: δ_C_ 168.59, 151.24, 147.23, 123.84, 121.62, 114.40, 112.32, 54.95.

### 3.6. Inhibition on Human Low Density Lipoprotein Oxidation Assay

#### 3.6.1. Isolation of LDL

Human blood was collected from healthy, non-smoking and 8-h fasting volunteers (Research Ethics No. UKM 1.5.3.5/244/SPP/NF-017). Nine mL of the blood were mixed with 1 mL of 3.8% sodium citrate solution and centrifuged at 2000× *g* for 20 min at 25 °C. The supernatant (plasma) was then collected for the LDL isolation [45]. The plasma (3.2 mL) was transferred into an 8.9-mL Optiseal^®^ tube and mixed with 0.8 mL of 60% iodixanol solution. This mixture was carefully layered with 4 mL of 12% iodixanol solution, followed by topping up to 8.9 mL with saline solution. The filled tube was then ultracentrifuged at 66,100 rpm (RCF_avg_ 299728× *g*, RCF_max_ 401268×g, *k* factor 40.9) for 3 h 10 min at 16 °C. After centrifugation, the mixture would be settled into four layers with very low density lipoprotein (VLDL) at the top part, followed by LDL, gap, and at the bottom of the tube was high density lipoprotein (HDL). The LDL, appearing as light orange layer, was carefully collected and placed into a new clean tube and quantified by using Protein Quantification Assay Kit from Sigma-Aldrich (St. Louis, MO, USA). Prior the assay, the density of the LDL was then adjusted to 200 µg protein per mL with PBS, pH 7.4.

#### 3.6.2. Inhibition of Copper-induced LDL Oxidation

The assay was performed based on Saputri and Jantan [44] with some modifications. About 378 µL of the diluted LDL (200 µg protein per mL) were mixed with 2 µL of sample and oxidized with 10 µM CuSO_4_. Ethanol and DMSO were used as blanks while probucol was used as positive control. For negative control, 380 µL of diluted LDL (200 µg/mL) and 10 µM CuSO_4_ were used. The mixture was vortexed and incubated at 37 °C for 5 h. After incubation, the tube was immediately put on ice for 10 min to stop the reaction. The oxidized LDL can be kept at −20 °C up to 2 days for MDA analysis (TBARS assay). The assay was performed in triplicate.

#### 3.6.3. MDA/TBARS Assay

The assay was performed following the provided manual of the assay kit with some modifications. Briefly, 40 µL of SDS solution were added into each treated-LDL, followed by 1 mL of TBA reagent solution, and well-mixed. The mixtures were then heated at 95 °C for 1 h. Then, the tubes were immediately placed on ice for 10 min to stop the reaction. The samples were centrifuged at 976× *g* at 25 °C for 15 min. The supernatants were then analyzed at 532 nm. The readings were corrected with their respective blank. The results were then plotted to the calibration curve of MDA standard solution. The IC_50_ value is considered as concentration of sample or drug which inhibits 50% of MDA production (Equation (1)):Percentage of inhibition = [1 − (MDA sample/MDA control)] × 100(1)

#### 3.6.4. Whole Blood Platelet Aggregation Assay

The assay used Whole-Blood Aggregometer^®^ (Chrono Log, Havertown, PA, USA) whereas the method has been optimized and established previously in our laboratory [44]. The blood was taken from healthy, non-smoker volunteers who have been 8-h fasting prior the blood collection (Research Ethics No. UKM 1.5.3.5/244/SPP/NF-017). Prior to the assay, the blood were added with 3.8% sodium citrate solution (9:1). The assay was started by mixing 492.5 µL of citrated blood with equal volume of saline solution and placing the magnetic bar in the cuvette. The mixture was incubated for 2 min in an incubation well (1st well). Then, 5 µL of sample were added and the reaction was incubated for 4 min. The cuvette was then transferred into sample well (2nd well). The electrode was put in and the machine was initiated for signal stabilization. After the signal was stable, 0.5 mM arachidonic acid (AA) or 10 μM adenosine diphosphate (ADP) was added and the reaction was run for 6 min. Total volume of the mixture was 1 mL. The whole assay was run at 37ºC. DMSO was used as a blank while acetyl salicylic acid (ASA) was used as a positive control. The resistivity, Ω, was recorded and the percentage of inhibition of aggregation was calculated as Equation (2):% inhibition = [(Ω blank − Ω sample)/Ω blank] × 100(2)

## 4. Conclusions

In conclusion, our findings suggested that *P. foetida* shows strong activity towards LDL oxidation and platelet aggregation which manifested potential activity against atherosclerosis. Flavonoids could be the compounds responsible for the plant extracts’ activities. Our results revealed that *P. foetida* might inhibit the oxidation of LDL by competitively decreasing copper-binding sites on the LDL that caused the oxidation process and at the same time neutralizing the radical species which were produced during the oxidation process. Some compounds in *P. foetida* extract might competitively inhibit platelet aggregation by docking with specific receptor(s) that are related to AA and ADP agonist pathways. Further studies need to be pursued to verify such mechanisms as well as to explore other possible antiatherosclerosis-related mechanisms of *P. foetida*.

## Figures and Tables

**Figure 1 molecules-24-01469-f001:**
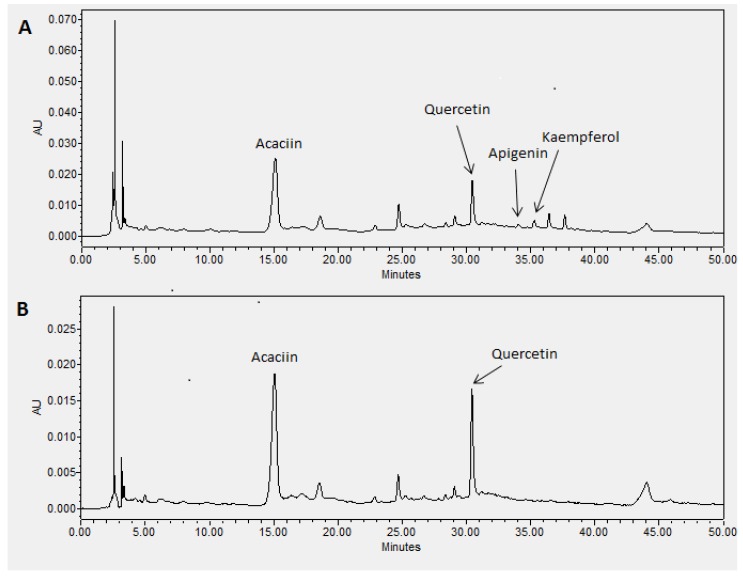
Representative RP-HPLC chromatogram of methanol extract of *Premna foetida* leaves at (**A**) 254 nm and (**B**) 360 nm for identification and quantification of acaciin (**1**), quercetin (**10**), apigenin (**8**) and kaempferol (**9**).

**Figure 2 molecules-24-01469-f002:**
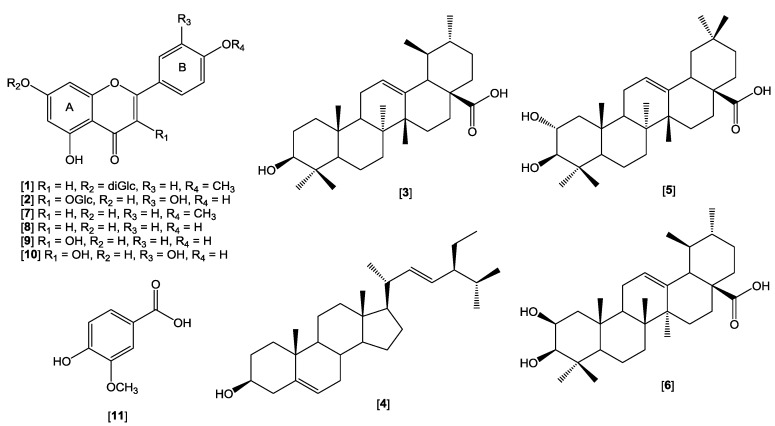
Isolated compounds from *Premna foetida* leaves.

**Figure 3 molecules-24-01469-f003:**
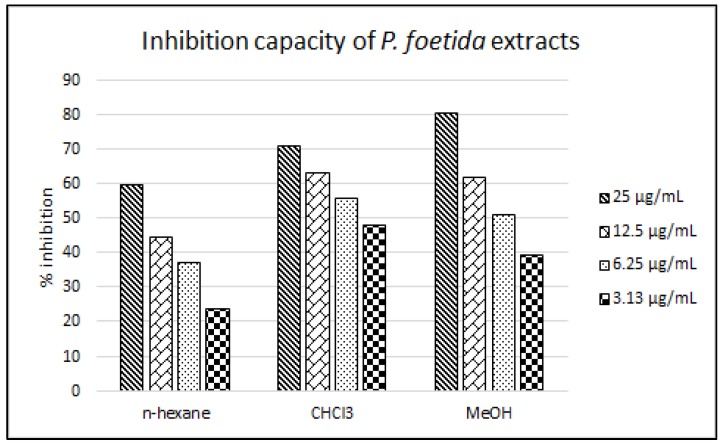
Trends in percent of inhibition of *Premna foetida* factions against LDL oxidation versus concentration in the following orders (bars, left to right: 25, 12.5, 6.25 and 3.13 µg/mL).

**Figure 4 molecules-24-01469-f004:**
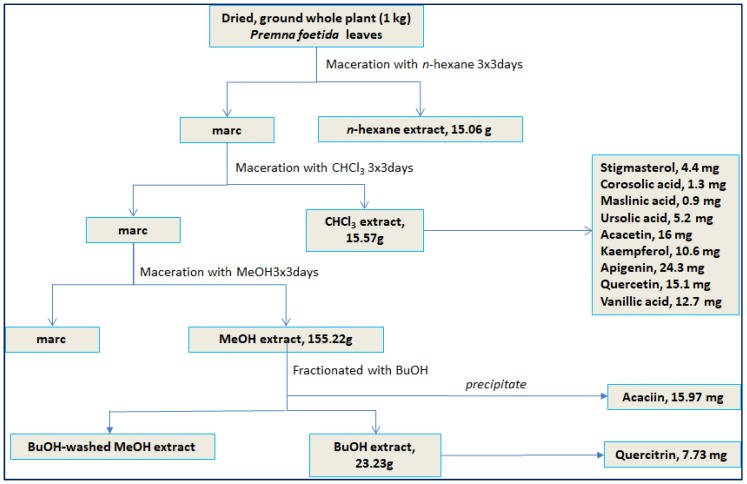
Isolation procedure of flavonoids and other compounds from *Premna foetida.*

**Table 1 molecules-24-01469-t001:** Validation parameters of RP-HPLC method for *Premna foetida* leaves extract analysis by using quercetin **10** as external standard.

Conc. (µg/mL)	% Recovery *	Intra-Day		Inter-Day	
		RT * (min)	Response **	RT * (min)	Response **
31.25	109.93 ± 6.01	30.04 ± 0.34	16700.7 ± 1.2	29.83 ± 0.09	16,611.3 ± 2.3
62.5	101.00 ± 2.03	29.79 ± 0.18	156,978.9 ± 2.3	29.69 ± 0.20	1568,90.3 ± 2.5
125	103.14 ± 1.86	29.94 ± 0.11	844,282.4 ± 0.8	29.89 ± 0.10	842,740.7 ± 0.4

Data are presented as * mean ± SD and ** mean ± RSD. Each concentration was injected in triplicate.

**Table 2 molecules-24-01469-t002:** Inhibitory activity of fractions and compounds from *Premna foetida* leaves against Cu^2+^-induced LDL oxidation.

Fractions/Compounds	Final Conc. (µg/mL)	Inhibition (%)	IC_50_, µg/mL (µM)
*n*-hexane fraction	25	59.80 ± 0.02	14.71 ± 0.11
	12.5	44.78 ± 1.38	
	6.25	37.26 ± 1.35	
	3.13	24.07 ± 0.75	
CHCl_3_ fraction	25	71.35 ± 0.31	3.95 ± 0.65
	12.5	63.16 ± 0.46	
	6.25	55.89 ± 1.85	
	3.13	48.09 ± 2.56	
MeOH fraction	25	80.55 ± 1.38	5.77 ± 0.08
	12.5	62.16 ± 0.25	
	6.25	51.05 ± 0.60	
	3.13	39.49 ± 0.17	
Linarin/Acaciin (**1**)	25	77.67 ± 0.32	7.41 ± 0.55
	12.5	66.16 ± 1.01	(12.51 ± 0.93)
	6.25	56.97 ± 0.26	
	3.13	49.20 ± 2.01	
Quercitrin (**2**)	25	70.84 ± 0.60	4.29 ± 0.13
	12.5	67.49 ± 0.28	(9.94 ± 0.29)
	6.25	53.14 ± 1.43	
	3.13	44.19 ± 0.38	
Ursolic acid (**3**)	25	54.95 ± 0.04	33.68 ± 1.28
	12.5	47.84 ± 1.56	(73.85 ± 2.80)
	6.25	31.26 ± 1.99	
	3.13	13.66 ± 1.94	
Stigmasterol (**4**)	25	56.95 ± 0.58	32.95 ± 0.92
	12.5	49.72 ± 0.14	(79.60 ± 2.23)
	6.25	36.51 ± 0.23	
	3.13	22.55 ± 0.35	
Acacetin (**7**)	25	73.61 ± 1.77	4.84 ± 0.30
	12.5	62.33 ± 2.03	(17.05 ± 1.06)
	6.25	53.49 ± 2.33	
	3.13	44.61 ± 0.62	
Apigenin (**8**)	25	77.77 ± 0.30	4.08 ± 0.04
	12.5	65.89 ± 0.70	(15.11 ± 0.16)
	6.25	4.44 ± 0.30	
	3.13	47.95 ±0.17	
Kaempferol (**9**)	25	83.55 ± 1.90	2.29 ± 0.13
	12.5	77.30 ± 0.34	(8.02 ± 9.44)
	6.25	66.23 ± 0.94	
	3.13	53.74 ± 1.17	
Quercetin (**10**)	25	100.00 ± 0.00	1.28 ± 0.04
	12.5	90.45 ± 0.55	(4.25 ± 0.12)
	6.25	78.27 ± 0.04	
	3.13	64.31 ± 0.26	
Vanillic acid (**11**)	25	63.74 ± 1.26	10.08 ± 0.22
	12.5	50.94 ± 0.04	(59.99 ± 1.28)
	6.25	42.66 ± 1.98	
	3.13	33.18 ± 0.59	
Probucol (positive control)	2.5	71.94 ± 1.55	0.21 ± 0.01
	1.25	65.50 ± 0.71	(0.40 ± 0.02)
	0.63	60.49 ± 0.64	
	0.31	56.15 ± 0.16	
	0.16	45.52 ± 0.37	

Data are presented as mean ± SD. The experiment was carried out in triplicate.

**Table 3 molecules-24-01469-t003:** Percent inhibitory activity of selected compounds from *Premna foetida* against AA- and ADP-induced platelet aggregation.

Compounds	Conc. (µg/mL)	AA (0.5 µM)		ADP (10 µM)	
		% Inhib.	IC_50_ (µM)	% Inhib.	IC_50_ (µM)
Apigenin (**8**)	100	100.0 ± 0.0	52.3 ± 1.3	99.4 ± 1.4	127.4 ± 2.3
	50	84.9 ± 1.2		63.2 ± 0.9	
	25	63.8 ± 0.7		33.0 ± 0.6	
	12.5	46.9 ± 1.4		9.3 ± 0.2	
Kaempferol (**9**)	100	74.4 ± 0.6	163.6 ± 5.0	65.2 ± 1.0	181.9 ± 4.6
	50	48.1 ± 1.9		51.5 ± 0.6	
	25	32.6 ± 0.5		31.7 ± 0.4	
	12.5	12.3 ± 0.1		11.1 ± 0.1	
Quercetin (**10**)	100	85.1 ± 0.6	124.9 ± 3.4	74.6 ± 1.1	173.2 ± 7.3
	50	55.0 ± 2.2		47.4 ± 0.7	
	25	37.3 ± 0.6		24.8 ± 0.5	
	12.5	14.1 ± 0.1		7.0 ± 0.2	
Aspirin (ASA)	25	100.0 ± 0.0	26.0 ± 1.3	44.8 ± 1.8	ND *
	12.5	85.2 ± 4.5			
	6.25	58.2 ± 2.3			
	3.13	36.5 ± 0.4			

Data are presented as mean ± SD. The experiment was done in triplicate. * ND: Not Determined

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
