# Peer review of "Inhibition of Human Platelet Aggregation and Low-Density Lipoprotein Oxidation by Premna foetida Extract and Its Major Compounds"

_molecules, 2019, doi:10.3390/molecules24081469_

Round 1

Reviewer 1 Report

Comments:

Inhibition of human platelet aggregation and low-density lipoprotein oxidation by  Premna foetida extract and its major compounds was reviewed. In general, the paper provides new and valuable information, being appropriated for publication in Molecules. However, some major points needed to be clarified.

1. Please explain why at low level (3.13 Î¼g/mL) of CHCl3 fraction had more inhibition activity to protect LDL from oxidation, but at high level (25 Î¼g/mL) of CHCl3 fraction decreased active as compared to MeOH fraction at two level of treatments? And what the major component in these fractions may cause the differences in preventing LDL from oxidation? 

2. Based on line 156-157 statement: If free hydroxyl group at C-3 and C-7 of C ring was giving superior effect than the substituted ones such as glycosylated or alkylated. Based on this concept applied to IC50 dataset, please explain why it seems that glycosylation of 7-OH (line 148) enhanced greater antioxidant effect in the flavone in the unchanged 4’-OH, such as acacetin and apigenin? Please re-consider this part of result statements.

3. Most tested flavonoids showed stronger inhibition towards AA - induced than the  

ADP- induced platelet aggregation with apigenin exhibited the strongest effect

while quercetin and kaempferol showed moderate activity. What the mechanism of a single hydroxyl substitution at C - 4’provided better activity than the di- substitution of hydroxyl at position C- 3’and C- 4’or a single hydroxyl at C- 4’, methylated B- ring and a non - hydroxylated planar C ring were potential to inhibit platelet function.

4. Wang et al. (2006) demonstrated that the H donation potential was quercetin > myricetin > morin > kaempferol, indicating that the presence of a 3′,4′-catechol moiety in the B ring correlated with high antioxidant activity. So, in line 22 indicated that hydroxyl group at C-3 may play important role in preventing the LDL oxidation, may not the main reason, please re-consideration. 

5. In materials and methods 3.6.2, 3.6.3, and 3.6.4 sections, please add all reagents the final concentration in molarity. 

6. In conclusion, line 368, inhibition the oxidation of LDL by chelating the transition metal that caused the oxidation process and so on. There is no evidence from the results to make this statement as one of conclusion. Please reconsideration for this statement.

7. Please carefully check the spalling with extra space.

Author Response

Inhibition of human platelet aggregation and low-density lipoprotein oxidation by  Premna foetida extract and its major compounds was reviewed. In general, the paper provides new and valuable information, being appropriated for publication in Molecules. However, some major points needed to be clarified.

1. Please explain why at low level (3.13 Î¼g/mL) of CHCl3 fraction had more inhibition activity to protect LDL from oxidation, but at high level (25 Î¼g/mL) of CHCl3 fraction decreased active as compared to MeOH fraction at two level of treatments? And what the major component in these fractions may cause the differences in preventing LDL from oxidation? 

2. Based on line 156-157 statement: If free hydroxyl group at C-3 and C-7 of C ring was giving superior effect than the substituted ones such as glycosylated or alkylated. Based on this concept applied to IC50 dataset, please explain why it seems that glycosylation of 7-OH (line 148) enhanced greater antioxidant effect in the flavone in the unchanged 4’-OH, such as acacetin and apigenin? Please re-consider this part of result statements.

3. Most tested flavonoids showed stronger inhibition towards AA - induced than the  

ADP- induced platelet aggregation with apigenin exhibited the strongest effect

while quercetin and kaempferol showed moderate activity. What the mechanism of a single hydroxyl substitution at C - 4’provided better activity than the di- substitution of hydroxyl at position C- 3’and C- 4’or a single hydroxyl at C- 4’, methylated B- ring and a non - hydroxylated planar C ring were potential to inhibit platelet function.

4. Wang et al. (2006) demonstrated that the H donation potential was quercetin > myricetin > morin > kaempferol, indicating that the presence of a 3′,4′-catechol moiety in the B ring correlated with high antioxidant activity. So, in line 22 indicated that hydroxyl group at C-3 may play important role in preventing the LDL oxidation, may not the main reason, please re-consideration. 

5. In materials and methods 3.6.2, 3.6.3, and 3.6.4 sections, please add all reagents the final concentration in molarity. 

6. In conclusion, line 368, inhibition the oxidation of LDL by chelating the transition metal that caused the oxidation process and so on. There is no evidence from the results to make this statement as one of conclusion. Please reconsideration for this statement.

7. Please carefully check the spalling with extra space.

Response to comment 1.

At high concentration (25 µg/mL), the MeOH fraction of P. foetida leaves showed the highest inhibitory activity (80.55 %) followed by CHCl3 and n-hexane fractions (71.35 and 59.80 %, respectively). However, at low concentration (3.13 µg/mL), the CHCl3 fraction produced more inhibitory activity than the MeOH fraction. Referring to Figure 3, a concentration-vs-inhibition relationship is observed and the slope trend exerted by the MeOH fraction is steeper than that of the CHCl3 fraction. This resuted in lower IC50 value of the CHCl3 fraction (3.95 µg/mL) than the MeOH fraction (5.77 µg/mL). Most of the flavonoids isolated in this study were obtained from the CHCl3 fraction (Figure 4). Thus it is deduced that the rich aglycon-flavonoid content present in the fraction might be responsible for such phenomenon. Changes have been made and highlighted in red in lines 110-118.

Response to comment 2.

Our findings were in agreement to previous studies [32,33] that reported the structure-activity relationship of some flavonoids towards LDL oxidation. Hence, the results indicated that (i) more free hydroxyl group would likely have higher antioxidant activity; (ii) two adjacent hydroxyl groups (catechol moiety) at B ring exerted superior antioxidant effect compared to the double bond and carbonyl groups at C ring; (iii) free hydroxyl group at C-3 and C-7 of C ring contributed to the antioxidant activity with free 3-OH provided better antioxidant capacity than the 7-OH. Glycosylated of 7-OH might improve the antioxidant activity of the flavone while glycosylation of 3-OH brought the opposite effect; and (iv) C2-C3 double bond and hydroxyl group at C-4’ also contributed to the antioxidant effect. Changes have been made and highlighted in red in lines 168-176.

Response to comment 3.

As mentioned in our discussion, our hypothesis is that it has to do with the receptor-binding (flavonoid-receptor interaction). Based on our results, it is concluded that the number of hydroxyl substitution does not  contribute significantly towards the binding. This is in agreement with  Beretz et [39] who found that the decrease of inhibitory effect of flavonoids towards platelet aggregation occurred with saturation of C2-C3 double bond,  lacking carbonyl carbon at C-4, glycosylation of C-3, and increasing number of hydroxyl substituents. However, we suggested that hydroxyl substitution at C-4'  in ring C of the flavonoid is necessary for the binding. Changes have been made and highlited in red in lines 205-215.

Response to comment 4.

Agreed. The catechol moiety contributes significantly towards antioxidant activity of the flavonoids, as many publications have reported this. We have made the changes in lines 21-23.  We have also included Wang et al. (2006) to support this as suggested (refer to lines 165-169).  However, our study focused on the isolated compounds which are mainly 3-OH group. Thus our structure analysis is based mainly on this particular 3-OH moiety.

Response to comment 5.

The final concentrations of all reagents have been changed to molarity. Changes are marked in red.

Response to comment 6.

We have made the changes as suggested. Refer to lines 388-391. The results revealed that P. foetida might inhibit the oxidatin of LDL by competitively decreasing copper-binding site on the LDL that caused the oxidation process and at the same time neutralizing the radical species which were produced during oxidation process. 

Response to comment 7.

The comments have been appropriately addressed.

Reviewer 2 Report

The manuscript entitled "Inhibition of Human Platelet Aggregation and Low-Density Lipoprotein Oxidation by Premna foetida Extract and Its Major Compounds" presents study of the methanol extract of P. foetida leaves (PFM) and its isolated compounds  for their ability to inhibit copper-mediated human low-density lipoprotein (LDL) oxidation and arachidonic acid (AA)- and adenosine diphosphate (ADP)-induced platelet aggregation. Most of the compounds are characterized by NMR and mass spectroscopy. The manuscript is well written and interesting. So, I would like to recommend for major revision. 

1. The author should revise the manuscript with the help of an english expert.

2.  The author have included positive controls in their study, but they should also include negative controls as well in each experiments.

3. It would be interesting if the authors include antiplatlet activities of the extracts as well, at least in the range of IC50 values.

4. The authors suggested that hydroxyl group at C-3 may play important role in protecting the LDL from oxidation. It would be interesting if the author could provide evidence based on molecular docking studies and the effective interaction environment between the active molecule and target enzyme.

5. It would be effective if the authors could support the inhibition at protein level by western blot analysis which provides evidence of the extent of the inhibition as well. Hence, i would recommend the author to perform some additional experiments.

Author Response

The manuscript entitled "Inhibition of Human Platelet Aggregation and Low-Density Lipoprotein Oxidation by Premna foetida Extract and Its Major Compounds" presents study of the methanol extract of P. foetida leaves (PFM) and its isolated compounds  for their ability to inhibit copper-mediated human low-density lipoprotein (LDL) oxidation and arachidonic acid (AA)- and adenosine diphosphate (ADP)-induced platelet aggregation. Most of the compounds are characterized by NMR and mass spectroscopy. The manuscript is well written and interesting. So, I would like to recommend for major revision. 

1. The author should revise the manuscript with the help of an english expert.

2.  The author have included positive controls in their study, but they should also include negative controls as well in each experiments.

3. It would be interesting if the authors include antiplatlet activities of the extracts as well, at least in the range of IC50 values.

4. The authors suggested that hydroxyl group at C-3 may play important role in protecting the LDL from oxidation. It would be interesting if the author could provide evidence based on molecular docking studies and the effective interaction environment between the active molecule and target enzyme.

5. It would be effective if the authors could support the inhibition at protein level by western blot analysis which provides evidence of the extent of the inhibition as well. Hence, i would recommend the author to perform some additional experiments.

Response to comment 1.

The manuscript has been revised to improve the English language.

Response to comment 2.

We have included negative controls in all the experiments. 

.For inhibition of copper-induced LDL oxidation assay, the negative control was 380 uL of diluted LDL (200 ug/mL) and 10 uM CuSO4 (line 356).    

For inhibition on whole blood platelet aggregation assay,  DMSO was used as a blank while acetyl salicylic acid was used as a positive control (lines 381-382).

Response to comment 3.

In the study design, antiplatelet activity was carried out only on pure compounds that have showed strong anti-LDL oxidation. Thus the extract was not subjected to the antiplatelet assay.

Response to comment 4.

Yes, we are considering to carry out molecular docking studies in future to look at effective interaction between the active compound and target protein. We have mentioned this in lines 213-215.  We proposed that the hydroxyl at C-3 may be involved in competitively inhibiting the copper to the binding site of the LDL, which is the histidine of the ApoB protein.

Response to comment 5.

Yes, Western blot analysis will provide more information on inhibition at the protein level. We are planning to include this aspect in future studies.

Reviewer 3 Report

Line 24: chnage to "exhibiting"

line 41: I prefer "the lipids"

line42: I suggest "attracts"

line 43: I suggest "induces"

line 47: The authors failed to discuss an article dealing with the MAO inhibitory activity of flavonoids (Inhibition of human monoamine oxidase: biological and molecular modeling studies on selected natural flavonoids. S. Carradori, M.C. Gidaro, A. Petzer, G. Costa, P. Guglielmi, P. Chimenti, S. Alcaro, J.P. Petzer. J. Agric. Food Chem. 2016, 64, 9004-9011.)

lines 83, 115, 173: chnage to "Data are"

lines 102 and 171: change to "percent"

line 111: corret "overrriden"

line 113: correct "Cu2+"

line 126: delete "and"

line 151: change to "seem"

line 182: change to "are damaged"

line 186: change to "blocked"

line 188: change to "inhibiting"

line 203: change to "promoted"

line 204: change to "exposed collagen"

lines 218, 219 and 220: change to "Data were"

line 227: correct "noe"

lines 227-231: the sentence is repeated twice

line 233: change to "were extracted"

lines 250, 252 and 257, 265: correct "CHCl3"

line 322: change to "were mixed"

lines 327 and 344: change rpm in g

line 335: change to "were mixed"

line 337: change to "were used"

lines 342 and 356 and 359: change to "were added"

line 346: change to "their respective"

line 371: change to "themselves into"

Author Response

Line 24: chnage to "exhibiting"

line 41: I prefer "the lipids"

line42: I suggest "attracts"

line 43: I suggest "induces"

line 47: The authors failed to discuss an article dealing with the MAO inhibitory activity of flavonoids (Inhibition of human monoamine oxidase: biological and molecular modeling studies on selected natural flavonoids. S. Carradori, M.C. Gidaro, A. Petzer, G. Costa, P. Guglielmi, P. Chimenti, S. Alcaro, J.P. Petzer. J. Agric. Food Chem. 2016, 64, 9004-9011.)

lines 83, 115, 173: chnage to "Data are"

lines 102 and 171: change to "percent"

line 111: corret "overrriden"

line 113: correct "Cu2+"

line 126: delete "and"

line 151: change to "seem"

line 182: change to "are damaged"

line 186: change to "blocked"

line 188: change to "inhibiting"

line 203: change to "promoted"

line 204: change to "exposed collagen"

lines 218, 219 and 220: change to "Data were"

line 227: correct "noe"

lines 227-231: the sentence is repeated twice

line 233: change to "were extracted"

lines 250, 252 and 257, 265: correct "CHCl3"

line 322: change to "were mixed"

lines 327 and 344: change rpm in g

line 335: change to "were mixed"

line 337: change to "were used"

lines 342 and 356 and 359: change to "were added"

line 346: change to "their respective"

line 371: change to "themselves into"

Response to comments.

All grammatical and spelling errors as highlighted, have been addressed appropriately. However, we opted not to include the reference in the introduction part since MAO enzyme is implied on psychiatric and neurological diseases while our study focused on the in vitro activities that are related to atherosclerosis pathogenesis.

Reviewer 4 Report

This is a study about antioxidation and inhibition on platelet aggregataion. Basically, this is only a preliminary results. Much more advanced studies are required. Regarding the flavonoids analysis, the results were not trustable. Most flavonoids are glycosides, and very few are aglycons. However, almost pure flavonoids are aglycon from the chromatogram. Need to be confirmed. The major problem, such kind of in vitro study does not mean anything. Some advanced desing are required.

Author Response

This is a study about antioxidation and inhibition on platelet aggregataion. Basically, this is only a preliminary results. Much more advanced studies are required. Regarding the flavonoids analysis, the results were not trustable. Most flavonoids are glycosides, and very few are aglycons. However, almost pure flavonoids are aglycon from the chromatogram. Need to be confirmed. The major problem, such kind of in vitro study does not mean anything. Some advanced desing are required.

Response: Yes, we agree that further studies are required to properly address and establish the mechanisms of the isolated flavonoids and other compounds in reducing the oxidation of LDL, especially the apo-B part, and platelet aggregation, including binding sites, antagonism/synergism etc.   Our report is the first phytochemical study on the isolation of six flavonoids, three triterpenoids, vallinilic acid and stigmasterol from Premna foetida.  The isolated  compounds were evaluated for their ability to inhibit copper-mediated human LDL oxidation and arachidonic acid- and adenosine diphosphate -induced platelet aggregation.  

In our study, few flavonoid aglycons were identified by HPLC but other peaks which might be representing other flavonoids including flavonoid glycosides were not identified.   The six  flavonoids isolated in this study were made up of glycons (acaciin, quercitrin) and aglycons (acacetin, apigenin, kaempferol, quercetin).

We are providing the NMR spectra of all the isolated compounds as supplementary documents.

The in vitro study on inhibition of the compounds on  human LDL oxidation and platelet aggregations provide useful information as platelets and oxLDL are  implicated in the pathogenesis of atheromateous lesion into the plaque which plays a key role in acute arterial thrombosis. The results of the present study will be used to pursue further studies to understand the underlying in vivo mechanisms. 

Round 2

Reviewer 2 Report

I am satisfied with the revised version of the manuscript. So, I recommend for publication. 

Author Response

Thank you for your positive response.

Reviewer 4 Report

Retention time at 15 mins is a big peak, but no description about this one. What is it?

What are the glycoside of the flavonoids? It seems only to see the aglycon. Basically, flavonoids glycosides are rich in the natural plant. Except some hydrolyzed treatment was used before analysis or extraction.

In vitro results are for reference. 

Author Response

1. Retention time at 15 mins is a big peak, but no description about this one. What is it?

Response to comment 1.

Unfortunately, we were not able to identify the peak at RT 15 mins. From our perspective and based on previous report on genus Premna, it is possibly either an iridoid glycoside or a phenylethanoid glycoside; since both groups are very common to be found in genus Premna. For our HPLC analysis, we only included peaks that are identifiable.

2. What are the glycoside of the flavonoids? It seems only to see the aglycon. Basically, flavonoids glycosides are rich in the natural plant. Except some hydrolyzed treatment was used before analysis or extraction.

Response to comment 2.

Yes, indeed, flavonoids can be found as glycosides but many studies also reported the isolation of aglycon flavonoids without hydrolysis process. In our previous published review [Ref. 25] on genus Premna, flavonoids as aglycons were reported in quite abundance where the isolation works did not necessarily require any hydrolysis step. In our opinion, flavonoids as aglycons are mostly distributed in the non-polar and semi polar fractions, depending on the methylation and hydroxylation substituents. Meanwhile, flavonoid glycosides are mostly isolated from the polar fraction.

In our HPLC analysis, we were not able to identify the flavonoid glycosides due to the unavailability of standard compounds.  However we managed to  isolate both aglycon and glycoside flavonoids from the plant extract. The aglycons are acacetin, apigenin, quercetin, and kaempferol. Meanwhile, the glycosides are acaciin/linarin (acacetin-7-O-diglucoside) and quercitrin (quercetin-3-β-O-rhamnoside).

3. In vitro results are for reference.

Response to comment 3 (In vitro results are for reference).

Did the reviewer meant the reference for in vitro results? In that case, the positive control used in our study showed similar result to a study by Saputri and Jantan (2011)  [Ref. 45].